# ProtChatGPT: Towards Understanding Proteins with Large Language Models

## Abstract

Protein research is crucial in various fundamental disciplines, but understanding their intricate structure-function relationships remains challenging. Recent Large Language Models (LLMs) have made significant strides in comprehending task-specific knowledge, suggesting the potential for ChatGPT-like systems specialized in protein to facilitate basic research. In this work, we introduce ProtChatGPT, which aims at learning and understanding protein structures via natural languages. ProtChatGPT enables users to upload proteins, ask questions, and engage in interactive conversations to produce comprehensive answers. The system comprises protein encoders, a Protein-Language Pertaining Transformer (PLP-former), a projection adapter, and an LLM. The protein first undergoes protein encoders and PLP-former to produce protein embeddings, which are then projected by the adapter to conform with the LLM. The LLM finally combines user questions with projected embeddings to generate informative answers. Experiments show that ProtChatGPT can produce promising responses to proteins and their corresponding questions. We hope that ProtChatGPT could form the basis for further exploration and application in protein research. Code will be publicly available.

## 1 Introduction

Proteins, as essential molecular entities for life, hold paramount significance in biological processes. The comprehensive understanding of protein structure and function is of utmost importance for advancing research in the realms of biology and biomedicine. However, traditional protein research normally involves labor-intensive laboratory experiments and extensive literature reviews, which could be time-consuming and require specialized expertise in protein.

Recently, Large Language Models (LLMs), *e.g.*, ChatGPT (Brown et al., 2020), have prevailed in Natural Language Processing (NLP) (Devlin et al., 2018; Raffel et al., 2020; Touvron et al., 2023; Chowdhery et al., 2022). With superior language understanding and logical reasoning capabilities, these models can perform various intricate linguistic tasks such as question and answering (Q&A). Since evolution through natural selection has spoken protein sequences as their "natural language", this intuitively motivates us to ride on LLMs' coattails and customize them into protein research based on large-scale biological corpora (*e.g.*, RCSB-PDB).

Empirically, with the capabilities of LLMs specialized in protein, researchers can potentially achieve *(1) Protein Understanding and Analysis* by simplifying the retrieval of crucial information (*e.g.*, structures, functions, interactions, mutations, and disease associations) about specific proteins for research; *(2) Customized Protein Design* by characterizing the patient's unique protein structures to discover targeted drugs and further verify expected functions for healthcare.

In this paper, we propose an AI-based protein chat system, named **ProtChatGPT**, to implement ChatGPT-like functionalities for the protein research field. ProtChatGPT works in a similar principle with natural language conversation systems. Users are allowed to upload protein 1D sequences or 3D structures (*e.g.*, fasta or pdb files) and pose diverse related questions. Then, ProtChatGPT produces comprehensive responses in an interactive manner based on the questions. In this way, researchers can intuitively acquire valuable insights and interactively delve into the complexities of diverse proteins. Specifically, ProtChatGPT consists of four components: protein encoders, a Protein-Language Pre-training Transformer (PLP-former), a projection adapter and an LLM, as shown in Figure 1. First, we employ two pre-trained protein encoders to embed the 1D (*i.e.*, by

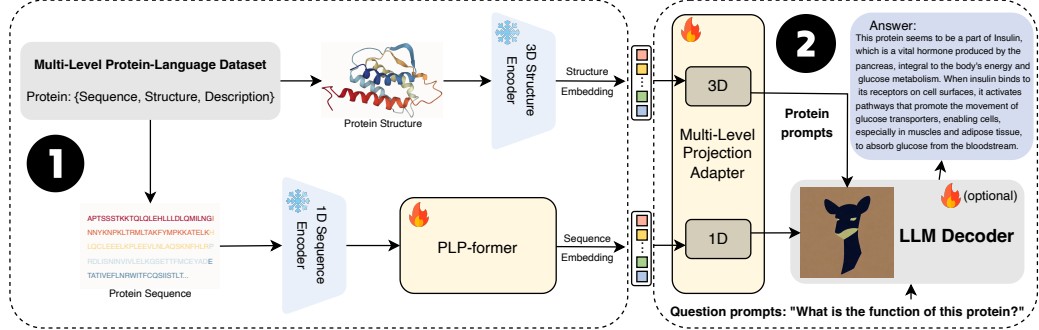

Figure 1: An overview of the ProtChatGPT framework. The training process consists of two stages: (1) protein-description representation learning stage, and (2) protein-to-text generative learning stage.

ESM-1b (Rives et al., 2021) ) and 3D structures (*i.e.*, by ESM-IF1 (Hsu et al., 2022)), respectively. Then, to align the protein and language modalities, we propose the PLP-former. PLP-former extracts features from the output of the protein encoder, and learns the protein representations that are most relevant to the text description. Third, we use an adapter as an information transmitter to convert protein embeddings into protein prompts that can be interpreted by the LLM. Finally, the LLM combines user questions (*i.e.*, question prompts) with the transmitted protein prompts to produce corresponding answers. We conduct experiments on protein understanding and design. Experimental results show the effectiveness of the proposed method. In summary, our contributions are as follows:

- We propose ProtChatGPT, an interactive ChatGPT-like system that engages Q&A for protein-related research, which significantly facilitates protein understanding and design.
- We introduce PLP-former, a transformer-based module that aligns the protein with its corresponding description.
- We propose a two-stage strategy that bootstraps protein-language pre-training from off-the-shelf pre-trained protein encoders and frozen large language models.
- We demonstrate ProtChatGPT's versatility and range of applications by deploying it to tasks of a rather distinct nature, including protein understanding and design.

## 2 RELATED WORK

**Protein Representation Learning.** Proteins are workhorses of the cell, which contain four distinct levels of structures carrying out their fundamental functions. Previous protein representation works seek to learn protein representations based on different levels of proteins. Considering protein sequences as language in life, several works (Madani et al., 2023; Notin et al., 2022) encode amino acid tokens using the Transformer model (Vaswani et al., 2017) to extract pairwise relationships among amino acids, and autoregressively recover protein sequences on extensive protein sequence databases. Alternatively, other sequence modeling methods(Lin et al., 2023; Meier et al., 2021; Rives et al., 2021; Rao et al., 2020; Vig et al., 2020) resort to use Masked Language Modeling (MLM) to develop attention patterns that correspond to the residue-residue contact map of the protein. Compared with sequence-based methods, structure-based methods (Gligorijević et al., 2021; OpenAI, 2023; Zhang et al., 2023) directly dictate protein functions and encode geometric information of proteins for topology-sensitive tasks such as molecule binding (Jin et al., 2021; Kong et al., 2022; Myung et al., 2022), protein interface analysis (Mahbub & Bayzid, 2022; Réau et al., 2023), and protein properties prediction (Zhang et al., 2022). In this paper, we aim to leverage these pre-trained Large Protein Models (LPMs) for high-quality embeddings without fine-tuning their network parameters.

**Large Language Models (LLMs).** Recently, Natural Language Processing (NLP) has witnessed significant advances due to the development of Large Language Models (LLMs) (Brown et al., 2020; Devlin et al., 2018) trained on an extensive, diverse corpus. Consequently, many multi-modal variants (Taylor et al., 2022; Alayrac et al., 2022; Jing et al., 2020; Wang et al., 2023; van Sonsbeek et al., 2023) based on LLMs have gained significant attention for the understanding of information in other modalities beyond text. For example, BLIP-2(Li et al., 2023) designs a Q-Former to align

the visual features from the frozen visual encoder with large language models. FROMAGe (Koh et al., 2023) freezes the LLM and visual encoders, and fine-tunes linear mapping layers to achieve cross-modality interactions. Similarly, MedVQA (van Sonsbeek et al., 2023) employs a multi-layer perceptron (MLP) network that maps the extracted visual features from a frozen vision encoder to a set of learnable tokens, which develops an open-ended VQA for diagnoses and treatment decisions. Galactica (Taylor et al., 2022) explicitly models the protein sequences and SMILES with scientific literature, and enables the model to explain the properties of the sequences. In this paper, we aim to adapt pre-trained general LLMs (*e.g.*, Vicuna (Chiang et al., 2023)) for protein-specific ChatGPT-like tasks, which aligns the protein features from LPMs with LLMs.

**Vision-Language Pretraining (VLP).** Data collected from different modalities generally offer distinct perspectives, frequently synergizing to yield a comprehensive understanding, enhancing the overall comprehension of the data. Vision-language pre-training (VLP) aims to learn multimodal foundation models, showing improved performance on various vision-and-language tasks (Radford et al., 2021a), Existing VLP methods can be roughly divided into *representation learning-based* and *generative learning-based*. *Representation learning-based* methods (Radford et al., 2021b; Jia et al., 2021; Yao et al., 2021; Li et al., 2022b; 2021) usually consider the image-text pairs as multi-modal views of the same semantics, and perform contrastive or multi-view learning for the alignment between multiple modalities. *Generative learning-based* methods (Li et al., 2019; Lu et al., 2019; Chen et al., 2020; Li et al., 2020; Zhang et al., 2021; Wang et al., 2022; Zeng et al., 2021; Bao et al., 2022) aim to reconstruct the corrupted text (image) with the assistance of visual (text) modality through MLM-like objectives. For example, SimVLM (Wang et al., 2021) introduces a single prefix language modeling (PrefixLM) objective for exploiting large-scale weak supervision in VLP. CoCa (Yu et al., 2022) further verifies the representation ability of autoregressive language modeling (AR) in the vision-language domain. In this paper, we consider protein as a specialized biological language that encodes and communicates biological information through its amino acid sequences and interactions. Thus inspired by existing VLP methods (Li et al., 2023; Zhu et al., 2023), we propose a Protein-Language Pre-training (PLP) framework to understand protein via natural language instructions.

## 3 METHODS

While demonstrating excellent performance in natural language tasks, LLM still lacks the capability to directly facilitate protein question-answering tasks due to the modality gap between protein structures and biomedical texts. As shown in Figure 1, in order to bridge this gap, we introduce a protein-language pre-training strategy with two stages: (1) protein-description representation learning stage and (2) protein-to-text generative learning stage. In the first pre-training stage, we enforce the PLP Transformer, a lightweight transformer with learnable query tokens, to extract features from a frozen protein 1D encoder, and learn the protein representation most relevant to the text description. In the second stage, we perform protein-to-text generative learning by connecting the output of the PLP Transformer as well as a supplementary frozen 3D encoder to a frozen LLM, and trains a multi-level adapter as an information bottleneck between two stages, such that its output protein representation can be interpreted by the LLMs.

### 3.1 ARCHITECTURE

As shown in Figure 1, ProtChatGPT consists of two pre-trained protein encoders, a trainable PLP-former, a trainable multi-level projection adapter and a pre-trained LLM decoder, working synergistically to provide protein-related insights.

**Multi-Level Protein Encoders** First, we use a pre-trained sequence encoder to extract protein sequence features. Specifically, given a protein sequence with $N$ amino acids, the encoder produces the corresponding sequence embedding $\boldsymbol{E}_{seq} \in \mathbb{R}^{N \times C_{seq}}$, where $C_{seq}$ is the number of embedding channels. In our implementation, we use ESM-1b (Rives et al., 2021) as the 1D sequence encoder, where $C_{seq} = 768$. Although ESM-1b is able to implicitly capture structural contact information, incorporating detailed 3D structures explicitly can be an effective way to model spatial interactions between residues. Therefore, we further propose to enhance the ESM-1b with a supplementary protein structure encoder ESM-IF1 (Hsu et al., 2022). Specifically, we select the feature from an

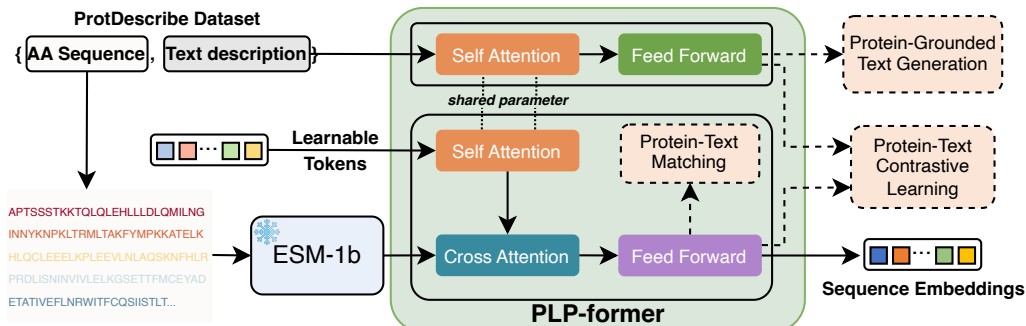

Figure 2: Illustrations of the PLP-Former and protein-language representation learning. PLP-Former consists of two transformer submodules with shared self-attention: (1) a text transformer that performs encoding and decoding of protein descriptions, and (2) a protein transformer that interacts with the frozen ESM-1b for sequence feature extraction. PLP-former is trained by jointly optimizing three pre-training objectives (dashed boxes) on sequence-description pairs.

intermediate layer as a complementary structure embedding $\boldsymbol{E}_{str} \in \mathbb{R}^{N \times C_{str}}$ where the number of embedding channels $C_{str} = 512$, indicating the geometric protein knowledge. By acquiring such multi-level property information, the quality of protein embeddings can be further improved, considering that the protein properties studied in ESM-1b and ESM-IF1 can correlate with each other. Note that, both the sequence and structure encoder are frozen for efficient training.

**Protein-Language Pretraining Transformer (PLP-former).** As mentioned before, the key challenge of transferring LLMs to protein research lies in the modality gap between protein structures and biomedical texts. Despite the strong language generation and zero-shot transfer abilities of LLMs, directly retraining them end-to-end for protein specialization appears to be impractical due to the massive number of parameters and data requirements. Another alternative is fine-tuning the pre-trained parameters, but this often leads to catastrophic forgetting. Considering this trade-off, we propose a Protein-Language Pre-training Transformer (PLP-former) for efficient cross-modal alignment between protein and text, while remaining LLMs frozen during the training. Following existing vision-language works (Li et al., 2023; Zhu et al., 2023; Dai et al., 2023), we use the PLP-former to extract protein-related features from a frozen protein sequence encoder.

As shown in Figure 2, the input of PLP-former consists of three parts: sequence embedding from ESM-1b, the corresponding description, and a set of learnable tokens. The learnable tokens $\boldsymbol{T}$ first perform mutual interactions via self-attention layers in the protein transformer. Specifically, the tokens $\boldsymbol{T}$ first acquire queries $\mathbf{Q}$, keys $\mathbf{K}$ and values $\mathbf{V}$ through three linear transformation matrices $\mathbf{W}_q$, $\mathbf{W}_k$ and $\mathbf{W}_v$. It can be formulated as:

$$\mathbf{Q} = \boldsymbol{E}_{str}^{l-1}\mathbf{W}_q, \mathbf{K} = \boldsymbol{E}_{seq}^{l-1}\mathbf{W}_k, \mathbf{V} = \boldsymbol{E}_{seq}^{l-1}\mathbf{W}_v. \tag{1}$$

Next, the attention map $w$ is computed by taking the dot product of Q and K as:

$$w = softmax(\frac{\mathbf{Q}\mathbf{K}^T}{\sqrt{d_k}}), \tag{2}$$

where $d_k$ represents the dimensionality of the keys, *softmax* is the softmax activation function. The refined tokens $\boldsymbol{T}'$ of the self-attention block can be written as:

$$\boldsymbol{T}' = \mathbf{W}_o \times (w \cdot \mathbf{V}), \tag{3}$$

where $\cdot$ represents the dot product, $\times$ means the matrix multiplication, and $\mathbf{W}_o$ is the projection matrix for output. Given the sequence features from ESM-1b as $\boldsymbol{E}_{seq}$, tokens then interact with $\boldsymbol{E}_{seq}$ through cross-attention layers, which can be formulated as:

$$\boldsymbol{T}'' = \mathbf{W}_o' \times Softmax\left(\frac{\boldsymbol{T}'\mathbf{W}_q'(\boldsymbol{E}_{seq}\mathbf{W}_k')^T}{\sqrt{d_k}}\right)\boldsymbol{E}_{seq}\mathbf{W}_v', \tag{4}$$

where $\boldsymbol{T}''$ represents the refined tokens after cross-attention, $\mathbf{W}'_q$, $\mathbf{W}'_k$, $\mathbf{W}'_v$ and $\mathbf{W}'_o$ are a new set of learning transformation matrices. Additionally, benefiting from the text transformer, tokens can further interact with the textual descriptions through the same self-attention layers. Finally, the PLP-former produces the output $\boldsymbol{E}_{seq}$ after a linear feed-forward layer (Vaswani et al., 2017). Depending on the pre-training task, we implement distinct attention masking strategies (Li et al., 2023) within the self-attention block to regulate the token-text interaction. The training details of PLP-former are given in Appendix A. In this way, PLP-Former can effectively select the most useful information for the LLM while removing irrelevant protein information. This reduces the burden of the LLM to learn protein-language alignment, thus mitigating the catastrophic forgetting problem.

Note that we apply PLP only to the sequence embeddings since the reported protein structures are much less than sequences. For example, there are 182K experimentally-determined structures in Protein Data Bank (PDB) (Berman et al., 2000) while 47M protein sequences in Pfam (Mistry et al., 2021). Thus we only use selected sequence-structure pairs during the second-stage training.

**Multi-Level Projection Adapter.** For the second protein-to-text generative learning stage, we further design a multi-level projection adapter to harvest the LLM's generative language capability. The adapter takes the pre-aligned sequence embedding $\boldsymbol{E}_{seq}$ from PLP-former and structure embedding $\boldsymbol{E}_{str}$ from ESM-IF1 as inputs, and acts as an information bottleneck to the LLM decoder, such that its output protein representation can be interpreted by the LLM. In practice, we use two individual Fully-Connected (FC) layers to linearly project the output protein embeddings into the same dimension as the question embedding of the LLM. They function as soft protein prompts that condition the LLM on protein representation from 1D and 3D levels.

**Large Language Models (LLMs).** Finally, the projected protein prompts are prepended to the question prompts (text embeddings of user questions) through concatenation. In implementation, we deploy the Vicuna-13b (Chiang et al., 2023) as our LLM decoder, which employs the Transformer decoder (Vaswani et al., 2017) to model the conditional generation probability $p_\theta(n_i|n_{<i})$ in the language model. Specifically, given the context vector of tokens $N_{i-1}$ (*i.e.*, protein embeddings along with user questions), the generated tokens $n_i$ (*i.e.*, answers) are computed as follows:

$$h_0 = N_{i-1}W_e + W_p, N_{i-1} = (n_1, n_2, ..., n_{i-1})$$
$$h_l = transformer(h_{l-1})(1 \le l \le m) \tag{5}$$
$$p(n_i|N_{i-1}) = softmax(h_m W_e^T)$$

where $m$ refers to the number of layers, $W_e$ denotes the token embedding matrix, $W_p$ represents the positional embedding matrix, and $transformer$ is a decoder layer consisting of a multi-head self-attention block and a position-wise feedforward network. During the second training stage, the generated tokens $n_i$ are replaced with publicly available descriptions for protein-to-text generative learning, while remaining empty in the testing stage, expecting ProtChatGPT to generate descriptive answers for the given protein and corresponding questions.

## 3.2  PROTEIN-LANGUAGE PRE-TRAINING DATASETS

In order to train ProtChatGPT, dedicated protein-specific training datasets are indispensable for our proposed two-stage training strategy. During the first stage, we adopt ProtDescribe dataset (Xu et al., 2023) to train the PLP-Transformer for protein-description representation learning. ProtDescribe dataset contains 553,052 aligned pairs of protein sequences and textual property descriptions such as protein names, functions, families, subcellular locations, etc. After the first stage, PLP-Transformer is able to acquire the protein 1D representation that is most pertinent to the textual description.

Nevertheless, despite covering the protein sequence positions, ProtDescribe de facto simply relies on textual descriptions to provide a rough indication of protein structural and functional similarity. It might be more straightforward to directly utilize structural information, especially considering that ESM-IF is specifically designed for protein geometric structures. Considering this problem, we resort to the RCSB-PDB Protein Description Dataset (Guo et al., 2023) which comprises 143,508 aligned pairs of protein 3D structures and the corresponding descriptions. We further expand this dataset with 1D sequences, enabling the joint training with our PLP-former. For every protein taken into account, we compile its 1D residue sequences, 3D atomic coordinates, along with corresponding scientific literature to train our multi-level adapter.

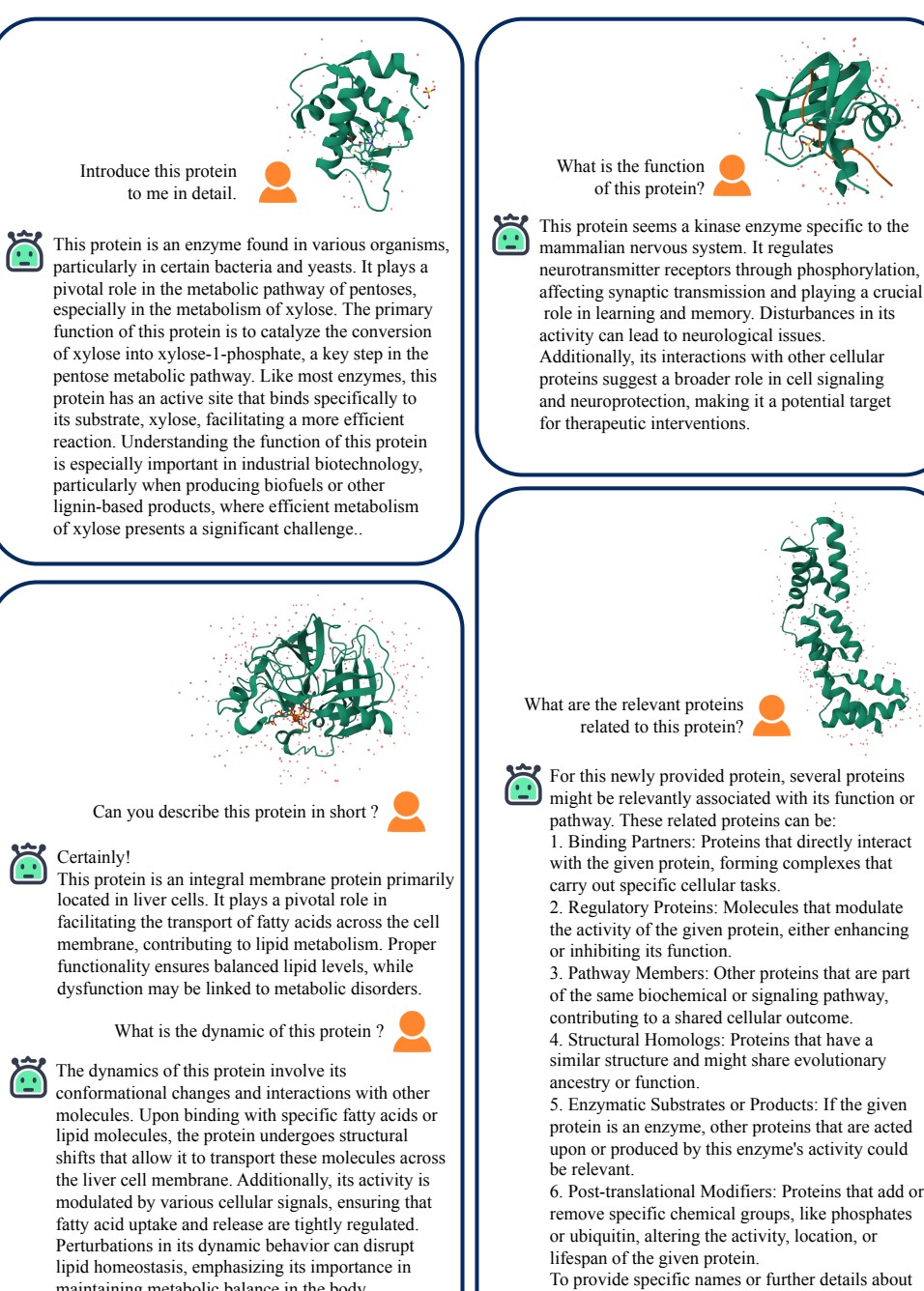

Figure 3: Dialogue examples of ProtChatGPT for protein understanding and analysis.

## 3.3  ALIGNED PROTEIN-TEXT GENERATION

To enhance the model training with protein-text pairs, we utilize a specialized token prompt, following the conversational format used in Vicuna-13b:

**Q**: $< Protein >< ProteinPrompts >< /Protein >< QuestionPrompts >$
**A**: $< Descriptions >$

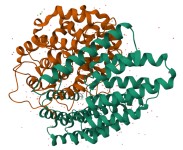

What diseases does this protein related to? 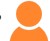

This protein is associated with a range of conditions due to its role in the life cycle of its parent microorganism. Related health issues include: mild infections, immune responses, digestive upsets, and allergic reactions.
In most cases, conditions linked to this protein are manageable and not life-threatening. Monitoring and basic medical care often lead to swift recovery.

How can I design a protein targeted on it? 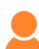

Given the protein's involvement in various conditions, potential therapeutic strategies include vaccine development, especially since it triggers immune responses. For digestive disturbances caused by this protein, probiotics might be beneficial to restore gut balance. Individuals showing allergic reactions to the protein could benefit from antihistamines to manage symptoms. Research can also focus on identifying compounds that inhibit the protein's function, reducing its pathogenic effects. Furthermore, developing diagnostic tests to detect this protein might aid in early treatment and better patient outcomes.

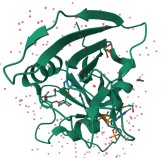

What physiological processes does the protein participate in? 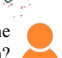

This protein plays a significant role in cell signaling, transmitting signals from the cell's surface to its interior, and mediating cellular responses to environmental stimuli. Additionally, it is involved in metabolism, assisting cells in converting nutrients into energy and other vital molecules. It may also contribute to the regulation of the cell cycle, ensuring timely cell division and maintaining tissue balance. The protein has a role in the immune response, either recognizing foreign pathogens or modulating the activity of immune cells. Lastly, it is potentially vital for protein synthesis, facilitating the translation of genetic information into functional proteins and overseeing their proper production and folding.

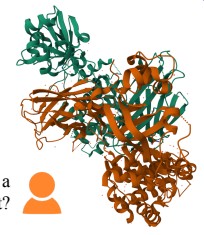

Can the protein serve as a drug target? 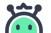

For the protein you provided, it could potentially serve as a drug target. This protein seems like Diphtheria-related toxin. It is a well-known virulence factor produced by certain strains of the bacterium Corynebacterium diphtheriae.
If the mutation affects the toxin's binding to host cells or its enzymatic activity, it could be a critical point of intervention. Drugs that target the mutant protein could inhibit its activity, rendering the toxin harmless, or prevent the toxin from binding to and entering host cells.
Moreover, understanding the structural and functional implications of the mutation could provide insights into designing more effective treatments or preventive strategies against diphtheria, especially if the mutant protein increases the virulence or alters the disease's presentation.

Are there any drugs currently available that target this protein? 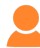

Given the context of the diphtheria toxin mutant protein you mentioned, as of my last training data cut-off in 2021, there are no specific drugs designed to target the diphtheria toxin mutant proteins directly.
For individuals who contract diphtheria, the primary treatment is diphtheria antitoxin, which neutralizes the toxin in the bloodstream. Antibiotics, like penicillin and erythromycin, are also used to treat the bacterial infection and prevent its spread to others.

What amino acid sequences should this antitoxin contain? 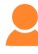

When discussing the functional regions of antibodies like diphtheria antitoxin, it's important to note that the exact amino acid sequences would depend on the specific antibody clone and the species from which the antitoxin is derived. However, I can provide hypothetical examples of peptide sequences that could be found in these regions:
Hinge Region: A possible sequence for flexibility might look like: PRO-THR-PRO-SER-PRO. Remember, these are just illustrative examples and would vary based on the specific antitoxin and the species it's derived from. Actual sequence determination would require experimental techniques such as sequencing or mass spectrometry.

Figure 4: Dialogue examples of ProtChatGPT for customized protein design.

where $< ProteinPrompts >$ represents the soft prompts that symbolize the multi-level embeddings aligned after the projection adapter. $< Protein >$ and $< /Protein >$ respectively represent the start and end symbols for protein embeddings. $< QuestionPrompts >$ represents the user questions that prompt the LLM to generate corresponding answers for the uploaded protein, such as "Describe the function of this protein". $< Descriptions >$ represents the generated answers, which have been substituted with publicly available protein descriptions during the second training stage. In the testing phase, they are left empty, anticipating ProtChatGPT to generate informative answers for the provided protein and associated questions.

## 4 EXPERIMENTS

### 4.1 IMPLEMENTATION DETAILS

For our training setup, we freeze both the sequence and structure protein encoders, as well as the LLM decoder, solely focusing on training the PLP-former (first stage) and the projection adapter (second stage). This two-stage strategy ensures that the pre-trained models retain their learned knowledge while fine-tuning the projection layer to align the protein embeddings with the LLM's requirements. For the first stage, we initialize PLP-former with the pre-trained weights of PubMedBERT (Gu et al., 2021), and randomly initialize the cross-attention layers. We use 32 learnable tokens with a dimension of 768, which is the same as the hidden dimension of the PLP-Former. We pre-train the PLP-former on the ProtDescribe dataset for 20K epochs with a batch size of 64. We use the AdamW (Loshchilov & Hutter, 2017) optimizer with $\beta_1 = 0.9$, $\beta_2 = 0.98$, and a weight decay of 0.05. We use a cosine learning rate decay with a peak learning rate of 1e-4, a minimum learning rate of 8e-5, and a linear warm-up of 5K iterations. For the second stage, we freeze the PLP-former and LLM, and train the projection adapter for 1K epochs with a batch size of 128. The minimum learning rate for the second stage is set as 5e-5. All experiments are performed on 4 NVIDIA A100 (80GB) GPUs, our model with ESM-1b and Vicuna-13b requires 5.5 days for the first stage and 2 days for the second stage.

### 4.2 RESULTS

#### 4.2.1 QUALITATIVE CASE STUDY

We first show some example conversations between the user and ProtChatGPT to indicate the promising results on both protein understanding and design tasks. For a fair comparison, we use the protein that does not appear in the training set of ProtChatGPT.

Figure 3 and Figure 4 respectively show some conversation examples of our ProtChatGPT related to protein understanding and design tasks. It can be observed that our system initially comprehends the meaning of the question well, providing accurate and logically consistent responses. Additionally, when presented with a protein, ProtChatGPT can provide explanations regarding questions associated with protein understanding. This implies that researchers can swiftly survey the structures, functions, and other relevant properties through ProtChatGPT. Coupled with this comprehension capability, further judgments on mutations and disease association can be made about specific proteins, leading to the potential of targeted protein design in healthcare.

#### 4.2.2 QUANTITATIVE COMPARISONS

To validate the effectiveness of our method, we further conducted several quantitative experiments. To fully showcase the capability of ProtChatGPT, we randomly selected 1,000 protein sequence-structure-description pairs from the RCSB-PDB Protein Description Dataset dataset to serve as the test set. Note that these testing protein pairs are not used during training for a fair comparison. We employed seven commonly used metrics in the image captioning and NLP domains to test the performance of ProtChatGPT. Detailed descriptions of these metrics can be found in Appendix B.

Table 1: Quantitative Comparisons on the proposed multi-level encoders and PLP-former. ↑ indicates that a higher value corresponds to better performance. The best performances are marked in **bold**.

| Metric
Variant | BLEU-1 ↑ | BLEU-4 ↑ | ROUGE-L ↑ | METEOR ↑ | CIDEr ↑ | SPICE ↑ | PubMed
BERTScore ↑ |
|---|---|---|---|---|---|---|---|
| *w/o structure* | 0.457 | 0.311 | 0.405 | 0.237 | 0.504 | 0.231 | 0.335 |
| *w/o PLP-former* | 0.581 | 0.352 | 0.463 | 0.270 | 0.572 | 0.276 | 0.421 |
| **ProtChatGPT** | **0.610** | **0.394** | **0.489** | **0.291** | **0.638** | **0.316** | **0.457** |

We first devised two variants to validate our contribution. (1) *w/o structure*: We removed the supplementary branch of the 3D structure encoder, relying solely on sequence information for learning. (2) *w/o PLP-former*: We omitted the PLP-former and directly aligned the 1D and 3D embeddings using the adapter. As a fair comparison, we give the LLM with the same user question as: "Describe this protein in short". It can be observed from Table 1 that both the structural encoder and

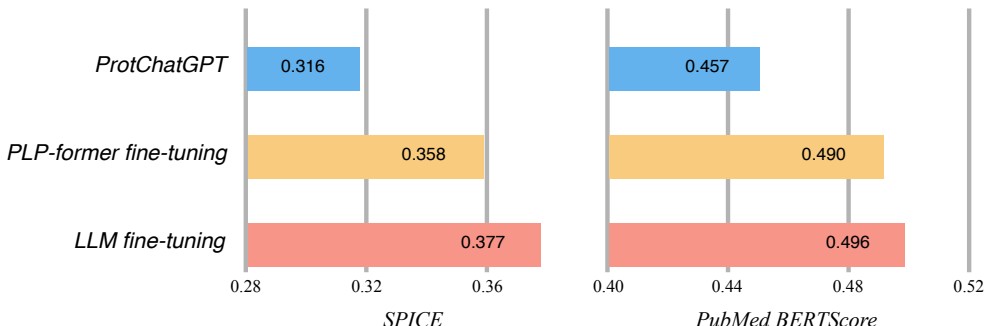

Figure 5: Comparison of fine-tuning of PLP-former and LLM decoder during the second-stage training. We compute the SPICE and PubMed BERTScore for semantic evaluation.

the PLP-former play indispensable roles in supplementation and alignment, respectively. Notably, in contrast to common metrics like BLEU and METEOR, SPICE and BERTScore pay more attention to deeper semantic information, rather than just lexical and syntactic alignment. Particularly, we replace the original Bert encoder with PubMedBERT (Gu et al., 2021), a biomedical description-specific encoder pre-trained on large-scale datasets. To some extent, this indicator can reflect the scientific validity of the generated responses in the biomedicine domain.

Furthermore, we modified the two-stage training strategy of PLP in an attempt to achieve better protein-specific dialogue capabilities. As shown in Figure 5, we independently fine-tune the LLM decoder and PLP-former in the second stage. The performance on two high-level semantic metrics SPICE and PubMed BERTScore indicate that further fine-tuning of both LLMs and PLP-former enhances the performance. Considering the computational cost, we adopt the fastest two-stage separate training strategy.

### 4.3 DISCUSSION AND FUTURE WORK

ProtChatGPT leverages the capabilities of LLMs for protein-specialized conversations. However, it inherits LLM's potential *language hallucination*. It is an indispensable concern especially when it relates to protein research and healthcare. Given an unknown protein, ProtChatGPT might produce certain descriptions that sound correct but lack proper scientific verification, possibly leading researchers astray. This issue might be alleviated by training the model with more high-quality, aligned protein-text pairs, or aligning with more advanced LLMs in the future. In this manner, rigorous data processing and selection strategies should be implemented to ensure the validity and reliability of the training data. One possible alternative is to further expand the dataset through structure-predicting models such as AlphaFold (Jumper et al., 2021). Feedback from domain experts is also important to refine the model. Combined with reinforcement or continual learning techniques, ProtChatGPT could keep improving the quality of its responses. Furthermore, training only one projection adapter might not provide enough capacity to learn extensive protein-text alignment. This issue could be alleviated by designing a more powerful adapter that further facilitates the interactions between sequence and structure embeddings. In future work, addressing these issues and refining ProtChatGPT is essential. With ongoing improvements and regular expert feedback, ProtChatGPT has the potential to become a trusted assistant in protein research, offering more valuable insights for further investigations.

### 5 CONCLUSION

In this paper, we introduce ProtChatGPT, an AI-based protein chat system to implement ChatGPT-like functionalities for the protein research field. ProtChatGPT marks the initial effort at bootstrapping Protein-Language Pre-training (PLP) from pre-trained LLMs for various protein-language tasks. PLP sits at the intersection between protein and language, which effectively and efficiently enables ProtChatGPT to harvest the off-the-shelf large models from both protein and natural language communities. Experiments suggest that ProtChatGPT holds potential for application in protein understanding and design. We hope this work can facilitate protein research and further inspire other scientific disciplines.

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

## A    DETAILS ON PROTEIN-LANGUAGE REPRESENTATION LEARNING

Obtaining queries that can extract informative protein representation regarding text is significant for protein-language alignment. To achieve this, we connect our PLP-former with the ESM-1b (Rives et al., 2021) model during the representation learning phase and train with the protein-language pairs. Following (Li et al., 2022a; 2023), we jointly train our model with three distinct pre-training tasks: Protein-Text Contrastive learning (PTC), Protein-grounded Text Generation (PTG), and Protein-Text Matching (PTM). Although these tasks utilize the same model structure and input format, they differ in the attention masking strategy applied between queries and text, thereby modulating their interaction.

**Protein-Text Contrastive Learning (PTC).**    For Protein-Text Contrastive Learning, by maximizing the mutual information, we aim to ensure the latent representation of protein and text are well-aligned. Specifically, given the query representation from the protein transformer $\boldsymbol{E}_{seq}$, we align it with the corresponding text embedding $t$. This is achieved by maximizing the similarity of positive pairs against those negative pairs where we directly use the embedding of $[cls]$ tokens from the text transformer as $t$. Given that the output of the protein transformer comprises multiple embeddings, we calculate the pairwise similarity between each query output and $t$. We then choose the highest value to represent the protein-text similarity. To prevent any information leakage, we utilize an unimodal self-attention mask that restricts direct interaction between queries and text.

**Protein-grounded Text Generation (PTG).**    The PTG task is designed to ensure that the learned queries can efficiently derive text-relevant information from the protein sequence. To accomplish this, we train the PLP-former to produce descriptions matching the respective protein sequences. Since the PLP-former prevents direct interaction between the frozen ESM-1b and text tokens, the data needed for description generation must first be garnered by the queries, ensuring efficient information extraction. We use a multimodal causal self-attention mask, to manage the interaction between queries and text. While queries can interact with one another, they cannot engage with the text tokens. Conversely, each text token can reference all queries as well as its preceding text tokens. Additionally, we substitute the $[CLS]$ token with a $[DEC]$ token at the beginning of the text sequence to indicate the decoding task.

**Protein-Text Matching (PTM).**    Protein-Text Matching task is leveraged for fine-grained protein-text representation alignment. This task is designed as a binary classification task where the model needs to determine if a given image-text pair aligns (positive) or misaligns (negative). We employ a bi-directional self-attention mask, allowing all queries and texts to mutually attend. As a result, the obtained query embeddings, $\boldsymbol{E}_{seq}$, encompass multimodal information. Each of these embeddings is then passed through a binary linear classifier to derive a logit, with the final matching score being the average of logits across all queries. For crafting informative negative pairs, we utilize the hard negative mining technique as described in (Li et al., 2021).

## B    DETAILS ON METRICS

In our implementation, we use seven different metrics on the proposed ProtChatGPT to verify the performance. These metrics not only focus on the low-level lexical and syntactic alignment but also on high-level semantic information.

**BLEU** (Papineni et al., 2002) (BiLingual Evaluation Understudy) serves as a crucial metric for assessing the quality of the machine-generated text, particularly in machine translation contexts. It quantifies the similarity between the candidate and reference text, yielding a score within the range of 0 to 1. A higher BLEU score indicates a closer match between the candidate and reference texts. BLEU is mathematically defined as follows:

$$p_n = \frac{\sum\limits_{C \in \{Candidates\}} \sum\limits_{n\text{-}gram \in C} Count_{clip}(n\text{-}gram)}{\sum\limits_{C' \in \{Candidates\}} \sum\limits_{n\text{-}gram' \in C'} Count(n\text{-}gram')} \tag{6}$$

$$BP = \begin{cases} 1 & \text{if } c > r \\ e^{(1-r/c)} & \text{if } c \leq r \end{cases} \tag{7}$$

$$BLEU = BP \cdot exp\left(\sum_{n=1}^{N} w_n log p_n\right) \tag{8}$$

where BP (Brevity Penalty) helps penalize overly short translations and $p_n$ represents the precision of n-grams, ranging from 1 to a predefined maximum.

**ROUGE-L** (Lin & Hovy, 2002) finds frequent use in the automatic evaluation of text summarization and machine translation. It calculates the longest common subsequence between the candidate and reference texts, prioritizing recall over precision. ROUGE-L can be expressed mathematically as:

$$R_{lcs} = \frac{LCS(X,Y)}{m} \tag{9}$$

$$P_{lcs} = \frac{LCS(X,Y)}{n} \tag{10}$$

$$\text{ROUGE-L} = \frac{(1+\beta^2)R_{lcs}P_{lcs}}{R_{lcs}+\beta^2 P_{lcs}} \tag{11}$$

In this context, $X$ represents the predicted text with a length of $n$. $Y$ represents the ground truth text with a length of $m$. $\beta$ is a hyperparameter used to adjust the emphasis on precision and recall. LCS calculates the length of the longest common subsequence, $R_{lcs}$ measures recall, and $P_{lcs}$ measures precision, respectively.

**METEOR** (Banerjee & Lavie, 2005) provides a comprehensive evaluation of machine-generated text by considering not only exact word matches but also synonyms and stemming. It combines precision, recall, and alignment factors to offer a holistic assessment. METEOR is mathematically represented as:

$$F = \frac{(\alpha^2+1)P}{R+\alpha P} \tag{12}$$

$$\text{Meteor} = (1 - Penalty) \cdot F \tag{13}$$

where $Penalty$ penalizes excessive word mismatches, $\alpha$ is a configurable parameter, $R$ and $P$ represents recall and precision respectively.

**CIDEr** (Vedantam et al., 2015) (Consensus-based Image Description Evaluation) primarily assesses the quality of image captions produced by automated systems. It places importance on consensus among multiple reference captions and emphasizes the inclusion of diverse descriptive words.

The mathematical formulation of CIDEr is given by:

$$CIDEr_n(c,S) = \frac{1}{M}\sum_{i=1}^{M}\frac{g^n(c) \cdot g^n(S_i)}{||g^n(c)|| \times ||g^n(S_i)||} \tag{14}$$

In the formula, $c$ represents the candidate text, $S$ denotes the set of reference texts, $n$ specifies the use of n-grams, $M$ represents the number of reference texts, and $g$ corresponds to the TF-IDF vector based on n-grams.

**SPICE** (Anderson et al., 2016) offers a metric designed to evaluate the semantic content of image captions, with a focus on their precision within generated captions.

SPICE's mathematical expression is as follows:

$$P(c,S) = \frac{|T(G(c)) \otimes T(G(S))|}{|T(G(c))|} \tag{15}$$

$$R(c, S) = \frac{|T(G(c)) \otimes T(G(S))|}{|T(G(S))|} \tag{16}$$

$$SPICE(c, S) = \frac{2 \cdot P(c, S) \cdot R(c, S)}{P(c, S) + R(c, S)} \tag{17}$$

where the binary matching operator $\otimes$ is the function that returns matching tuples in two scene graphs, $P$ represents the precision of semantic propositions and $R$ signifies the recall of semantic propositions, respectively.

**BertScore** (Zhang et al., 2019) is a metric that leverages contextual embeddings from BERT models to assess the quality of machine-generated text. It measures the similarity between the candidate text and the reference text using contextual embeddings.

**PubMed BERTScore** comes from the classical **BertScore** (Zhang et al., 2019), which is a metric that leverages contextual embeddings from BERT models to assess the quality of machine-generated text. It measures the similarity between the candidate text and the reference text using contextual embeddings. In our implementation, to better assess the quality of ProtChatGPT in generating protein-related descriptions, we further replaced the encoder with the encoder of PubMedBERT (Gu et al., 2021). PubMedBERT is the latest BERT (Devlin et al., 2018) model pre-trained on the biomedical corpus, which outperformed BioBERT on the BLURB (Gu et al., 2021) (Biomedical Language Understanding and Reasoning Benchmark).

