# OpenReview forum: "ProtChatGPT: Towards Understanding Proteins with Large Language Models"
_ICLR.cc/2024/Conference — ICLR 2024 Conference Withdrawn Submission_

### Official Review · Reviewer_o37r · 2023-10-26

**Soundness:** 3 good
**Presentation:** 3 good
**Contribution:** 3 good
**Rating:** 6
**Confidence:** 4

**Summary:**

This paper introduces an interesting ChatGPT-like system named ProtChatGPT, which combines a large language model (text) with a protein language model (protein sequence or structure) through a designed PLP-former and projection adapter.

The primary goal of this paper is to enhance ChatGPT's capability to analyze protein sequences/structures and respond to questions related to proteins. The motivation behind this research is substantial, and the proposed methodology offers valuable insights. Moreover, the experimental results provide compelling evidence that ProtChatGPT can generate promising responses, potentially inspiring future advances in protein research

**Strengths:**

This paper introduces ProtChatGPT, a ChatGPT-like system tailored for applications in the field of protein research. The primary innovation of ProtChatGPT lies in its capacity to process protein sequences or structures and provide informative responses to relevant queries. The proposed PLP-former serves to align protein data with corresponding descriptions, thus generating coherent protein prompts. Furthermore, the LLM effectively combines user questions with the prompts to generate relevant answers.

In summary, this paper provides a well-organized presentation of an innovative concept. I find the idea presented in this paper quite interesting. The preliminary results show promise, and it's evident that ProtChatGPT has significant potential to make valuable contributions to the field of bioinformatics and related research areas.

**Weaknesses:**

The experimental section requires further elaboration, with an emphasis on in-depth result analysis rather than a mere display of examples. Figures 3 and 4 are quite extensive and should be presented more concisely.

As noted by the author in the "4.3 Discussion and Future Work" section, ProtChatGPT shows the potential to generate descriptions that, although seemingly plausible, might lack comprehensive scientific validation, especially for proteins with unknown functions. I am particularly intrigued by how ProtChatGPT performs when presented with homologous proteins. If space allows, I suggest the author explore the possibility of including comparisons and analyses of homologous proteins that possess similar sequences or structures but manifest significantly different functions. Such an experiment could further enhance the validation of ProtChatGPT's utility.

**Questions:**

As a researcher in the field of bioinformatics AI, I find this work intriguing. It would be immensely beneficial if you could offer a demonstration website that allows reviewers to engage with practical examples. While your proposed methodology appears sound and the experimental results are compelling, firsthand interaction with ProtChatGPT during the review process would be the ideal way to underscore the value and versatility of your research.

I would be more inclined to give a higher rating if I had the opportunity to experience ProtChatGPT for myself.

---

### Official Review · Reviewer_uU9h · 2023-10-28

**Soundness:** 2 fair
**Presentation:** 2 fair
**Contribution:** 2 fair
**Rating:** 5
**Confidence:** 3

**Summary:**

This paper presents a which leverages large language model to understand the functionality and the implicit representation of proteins from  both structure and sequence modalities with joint embedding. Additionally, this paper presents PLP former, a Blip2-like module with shared self-attention of text and protein sequence transformer.

**Strengths:**

1. This paper leverages the strength of LLM and aims to provide a pipeline that could interactively understand the protein from both structure and sequence perspective.
2. It presents a Blip2-like module and shows that this module, on top of the pipeline, improves the overall performance on multiple metrics.
3. This outcome of this paper could be helpful in propagating protein understanding to broader audiences.
4. The paper is clear on the model architecture and training pipeline.

**Weaknesses:**

1. This paper lacks experiment supports from baseline. The only results are presented in Table 1 which is conducted within this pipeline. It would be more subjective to include the results from other relevant papers.
2. The main concern is lack of novelty. Most of the parts are pieced together from existing works (ESM1b, ESMIF1, Vicuna 13b etc.) In addition, the work flow is similar to https://www.techrxiv.org/articles/preprint/ProteinChat_Towards_Achieving_ChatGPT-Like_Functionalities_on_Protein_3D_Structures/23120606 with addition of protein sequence and PLP former. However, the idea of PLP former draws clear analogy from Blip2 with little innovation.
3. This paper currently does not have code and cannot run sanity checks.

**Questions:**

1. ESM-1B (650M) + ESM-IF1 (142M) + Vicuna 13B have very large number of parameters, and for long sequences and structures, it is very likely to cause OOM. How this problem is handled during the training?
2. How does the model perform in setting ProtChatGPT w/o sequence?
3. Have you tried other protein encoders, for example GearNet for protein structures?
4. ProtDescribe is using uniprot functionalities and RCSB-PDB Protein Description Dataset is using PubMed abstract. What is the textual input in your case?

---

### Official Review · Reviewer_iHd4 · 2023-10-31

**Soundness:** 3 good
**Presentation:** 3 good
**Contribution:** 3 good
**Rating:** 6
**Confidence:** 4

**Summary:**

This work presents a model for a chat agent trained on protein question answering. The model encodes input protein sequence and structure information by leveraging powerful protein representation models, ESM-1b and ESM-IF1, respectively. The model employs a so-called “Protein-Language Pretraining” (PLP) module, analogous to Visual-Language Pretraining to align the modalities of protein sequence and natural language description of a protein. Structure and sequence embeddings are combined and projected into a shared space with a LLM decoder to produce natural language responses to prompts about the input protein.

**Strengths:**

This model is able to combine multiple modalities of information related to proteins, sequence, structure, and descriptions. The use of the PLP-former is able to refine pre-trained sequence embedding features for the task of text description generation without losing the power of the original features.

Combines sequence and structure features via the use of the “multi-level projection adapter.” Since encoding models (such as the ESM models used here, or others) are now easily accessible and widely used for representing proteins, making use of both is the way to go. The empirical results in Table 1 support this, showing the performance degradation when structure is not incorporated.

**Weaknesses:**

There are several points of ambiguity with the dataset and modeling (see the Questions section for details).

Although several NLP-based quantitative evaluation metrics are included in evaluation, these are likely insufficient for this task where it is not clear if the chat responses are correct. Some analysis, even if a case study of a few proteins, would strengthen the submission. E.g. Do responses have mutually exclusive functions listed, etc? For a work that aims to serve drug development or protein design, it is not sufficient to have the grammar and appearance of a correct response, but to also be factually correct. The SPICE and PubMedBERT scores move closer to such an evaluation, but they are not sufficient to “trust” the output of such a model yet. The authors rightfully list these concerns in their discussion. Some investigation could be done presently to highlight and quantify these limitations

**Questions:**

Q1] The text description in the dataset are not clear. Some points are listed in the first paragraph of 3.2, but is not comprehensive. Where was the text taken from? For example, when subcellular location is mentioned, how is this represented? E.g. literature excerpts, GO terms, etc? More detail about the training descriptions would be helpful.

Q2] “which comprises 143,508 aligned pairs of protein 3D structures and the corresponding description” What is mean here by “aligned”? Does this refer to the structure-description pairing, an alignment of proteins, or something else?

Q3] The last component of the input data that is mentioned is “corresponding scientific literature.” What is meant by this? How is this literature chosen and used in the model?

Q4] How were training and test sets split? If random split, what was the overlap, sequence/description/family/function similarity across splits?


Minor comments and additional suggestions

For clarity, consider adding which of the VLP types your approach falls into (i.e. representation vs generative)

In the second paragraph of Section 1 RCSB-PDB is introduced without explaining what it is. The sentence containing this reference is also very unclear. I suggest re-framing it to make clear that you are making a connection between language and the evolutionary information contained in protein sequences. “Evolution” does not have a “spoken language” as such.

It is not clear how the learnable tokens are initialized/where they are coming from and their role. Why use 32 etc?

Notation slightly inconsistent, especially in equation (5) and accompanying description where matrices are vectors no longer in bold font.

---

### Official Review · Reviewer_MnUj · 2023-11-03

**Soundness:** 2 fair
**Presentation:** 2 fair
**Contribution:** 2 fair
**Rating:** 3
**Confidence:** 4

**Summary:**

The paper presents a protein chatbot that aims to converse about proteins. The method relies on two trainable components (the PLP-former and projection adapter) that take as input embeddings from pretrained protein models and transform them into a representation suitable for a Vicuna 13B LLM.

While this is clearly an interesting area of research, I found the approach not well evaluated and thus the paper not that convincing. The novelty of the method is also modest.

**Strengths:**

-The paper touches an important problem of connecting proteins to the text modality in an aspirational conversational system.

**Weaknesses:**

-The evaluations/benchmarks presented are not that convincing. The only quantitative evaluation presented is on the RCSB-PDB Protein Description Dataset and the only results shown are that of various ablations of the authors' proposed system. Only some standard n-gram matching metrics are presented. There is no human evaluation or more factuality oriented metrics that would attest to the actually utility of the system.

-The writing could be more rigorous, particularly in describing the datasets and how they are used to train the model (Section 3.2). This would help give the reader more intuition on what the output of the PLP-former is supposed to represent.

-The name ProtChatGPT is not appropriate. ChatGPT is a commercial system and none of the components of the proposed method use it.

-The novelty of the method is modest, mostly an aggregation of existing techniques.

Nits:
-The citation to ChatGPT on the first page is incorrect.

**Questions:**

It would be great to have some details/examples on Section 3.2, e.g. how the datasets are used to train the different modules (e.g. what the inputs and outputs are etc.), maybe an example would be useful.